# Simultaneous Generation of Complex Structured Curve Beam

**DOI:** 10.3390/nano9010087

**Published:** 2019-01-11

**Authors:** Jun Wu, Xinquan Tang, Jun Xia

**Affiliations:** Joint International Research Laboratory of Information Display and Visualization, School of Electronic Science and Engineering, Southeast University, Nanjing 210096, China; 220181301@seu.edu.cn (X.T.); xiajun@seu.edu.cn (J.X.)

**Keywords:** beam shaping, computer holography, spatial light modulation

## Abstract

At present, people are using holographic technologies to shape complex optical beams for both fundamental research and practical applications. However, most of the reported works are focusing on the generation of a single beam pattern based on the computer-generated hologram (CGH). In this paper, we present a method for simultaneously shaping the multiple beam lattice where the intensity and phase of each individual beam can be prescribed along an arbitrary geometric curve. The CGH that is responsible for each individual beam is calculated by using the holographic beam shaping technique, afterwards all the CGHs are multiplexed and encoded into one phase-only hologram by adding respective linear phase grating such that different curves are appeared in different positions of the focal regions. We experimentally prove that the simultaneous generation of multiple beams can be readily achieved. The generated beams are especially useful for applications such as multitasking micro-machining and optical trapping.

## 1. Introduction

Optical vortices are a type of special complex light field which has many important properties. In 1979, Vaughan et al. conducted a series of studies on Laguerre-Gaussian (LG) beams using interferometric methods, and experimentally verified the existence of optical vortices [1]. In 1989, Coullet et al. found that there are many similar states in the resonant cavity when studying laser resonators with large Fresnel Numbers. The concept of “optical vortices” is provided for the first time [2]. In 1992, Allen et al. studied the characteristics of light waves in the condition of near-axial propagation and non-axial propagation, and found that optical vortices with phase factors possess orbital angular momentum (OAM) [3]. The OAM provides a valuable theoretical basis for optical vortices applications in many fields, such as optical micro manipulation, quantum communication, optical imaging, optical measurement, and so on [4,5,6,7,8]. Recently, Liu et al. studied manipulation of the ellipsoidal micro-particles by the femtosecond vortex tweezers both experimentally and theoretically [5]. With the increasing understanding and mastery of optical vortex characteristics, researchers are paying more attention on the generation of optical vortices. Carmel Rotschild et al. generated a vortex curve through a spiral phase plate [9]. But the usage of such a device is restricted by the high accuracy requirement and complex configuration. Y. Izdebskaya et al. studied diffraction of a Gaussian beam with a system of successively located optical wedges, and generated higher-order optical vortices [10]. D. Sarenac et al. described a parallel multiplexing technique that produced a beam consisting of a lattice of OAM states coupled to a two-level system through N sets of Lattice of Optical Vortices (LOV) prism pairs [11]. Optical vortices can also be generated by loading a designed phase-only hologram into the spatial light modulator (SLM) and meanwhile irradiating the SLM with lasers. However, the above-mentioned technique can only produce a single optical pattern in the focal plane. In 2016, Deng et al. proposed an approach for creating three-dimensional (3D) multifocal vortices arrays [12]. The position, orbital angular momentum states, number and diameter of each beam can be freely modulated, but the technology simply duplicates the same pattern on different regions, restricting its application in more complicated beam circumstances.

In 2013, Rodrigo et al. developed a technique to allow the generation of high intensity gradient (HIG) beams whose phase and intensity are prescribed based on the calculation of a computer-generated hologram (CGH) [13]. They experimentally proved that the beams in distinct 3D geometries can be shaped. They also showed that a freestyle laser trap, including high-intensity and phase gradient forces, was able to confine multiple particles and drive their motion [14].

It is critically important both in theory and applications to produce optical curve beams of different tunable designed patterns in focal regions. In this paper, we both theoretically and experimentally demonstrate the method for simultaneous generation of multiple beams using the modified holographic beam shaping technique, where the single phase-only CGH is calculated by spatially multiplexing and encoding multiple complex CGHs responsible for the shaping of different specific amplitude and phase distributions. In addition, the position of each curve pattern can be individually modulated and controlled along transversal as well as axial dimensions. Such new light beams are expected to expand the application field of optical vortices and are also potentially useful in the realization of super-performance multitasking optical applications.

## 2. Materials and Methods

Figure 1b shows the scheme of a holographic beam shaping technique which allows designing complex beams whose intensity and phase distribution follow a prescribed curve. The computer-generated hologram (CGH) of a 2D ring curve beam is shown in Figure 1a. Specifically, in order to generate a desired focal beam, the complex amplitude of the incident plane is given by [15],
(1)$$G\left(x,y\right)=\int_{0}^{2\pi }\varphi \left(x,y,t\right)\phi \left(x,y,t\right)\sqrt{{\left[x_{0}^{\prime }\left(t\right)\right]}^{2}+{\left[y_{0}^{\prime }\left(t\right)\right]}^{2}}\mathrm{dt}$$
where the terms ϕ(x,y,t) and φ(x,y,t) are determined by
(2)$$\varphi \left(x,y,t\right)=\exp\left\{i\pi \frac{{\left[x-x_{0}\left(t\right)\right]}^{2}+{\left[y-y_{0}\left(t\right)\right]}^{2}}{{\lambda f}_{0}^{2}}z_{0}\left(t\right)\right\}$$
and
(3)$$\phi \left(x,y,t\right)=\exp\left\{\frac{i}{\omega_{0}^{2}}\left[{\mathrm{yx}}_{0}\left(t\right)-{\mathrm{xy}}_{0}\left(t\right)\right]+\frac{i\sigma }{\omega_{0}^{2}}\int_{0}^{t}\left[x_{0}\left(\tau \right)y_{0}^{\prime }\left(\tau \right)-y_{0}\left(\tau \right)x_{0}^{\prime }\left(\tau \right)\right]d\tau \right\}$$
where [x_0_(t), y_0_(t), z_0_(t)] represents the prescribed curve in the Cartesian coordinate with t∈[0,2π]. f_0_ and λ refer to the focal length of the Fourier lens and the wavelength, respectively.

Equation (1) enables calculating the incident complex field (namely, complex CGH) that can shape a structurally stable focal beam with special intensity distribution and phase gradient (helical phase along the curve). We first consider a ring curve x_0_(t) = Rcos(t) y_0_(t) = Rsin(t) and demonstrate the performance of this technique by simulation. The intensity distribution of the resulting beam is displayed in Figure 1b. The phase distribution of the ring is well defined along curves under the topological charge of m = 1 (see Figure 1f). Besides this, we consider other three shapes: an Archimedean spiral, a trefoil-knotted curve and a square curve. The topological charge is m = 1 for all the curves. For the sake of clarity, the corresponding curve parametric expressions are provided in Table 1. The intensity distribution and the phase distribution of the resulting beam is displayed in Figure 1b–i.

In order to multiplex various curve beams partially separated in the far field, each complex CGH calculated by the Equation (1) must be encoded with a unique carrier frequency. This can be achieved by adding a linear phase grating to the hologram of each beam. Linear gratings in combination with spatial filters are commonly used to isolate the first diffraction order from undesired zero and higher diffraction orders. The transfer function of a linear phase grating is given as
(4)$$\varphi_{i}\left(x,y\right)={\mathrm{kz}}_{i}\sqrt{1-\frac{x^{2}}{f_{0}^{2}}-\frac{y^{2}}{f_{0}^{2}}}+k\left(\frac{{\mathrm{xu}}_{i}}{f_{0}}+\frac{{\mathrm{yv}}_{i}}{f_{0}}\right)$$
where u_i_ and v_i_ are the spatial coordinates of the generated beam in the far field, achieved with a Fourier lens of focal length f_0_. K = 2π/λ is the wave number. z_i_ is the axial shifted displacement away from the focal plane (Fourier plane). In order to generate curve beams simultaneously, the expressions of the final complex CGH need to be added together by
(5)$$H\left(x,y\right)=\sum_{i=1}^{n}G_{i}\left(x,y\right)\exp\left[{i\varphi }_{i}\left(x,y\right)\right]$$

The conventional methods of shaping optical beams are generally suitable on a single focal plane. In our method, the relative position of the resulted beams can be flexibly adjusted along the z-direction. Before the experiment, we predicted the proposed method through the simulation and generated the phase-only hologram for subsequent experiments. Four different curve shapes: a ring curve, an Archimedean spiral, a trefoil-knotted curve and a square curve were considered. The topological charge is m = 5 for all the curves. These four curve beams are focused simultaneously on the focal plane (z = 0). Simulation results are shown in Figure 2. The relative position and topological charge of the four kinds of curve beams complex structured curve beams can be controlled by adjusting the parameters of (z_i_, u_i_, v_i_ and m) in the CGH calculation.

We have designed two structures to prove that the above four curve beams can be generated on different focusing regions along the optical axis (z-direction). The ring curve and the square curve are still focused on the focal plane (z = 0) while the Archimedean spiral and the trefoil-knotted curves have axially shifted to the plane of z = −0.05 m and z = 0.05 m, respectively. Figure 3 and Figure 4 show the two different structures. Intensity distribution of the beams projected in the focal plane (z = 0), seen in (c), and the Archimedean spiral and the trefoil-knotted curve are focused on the z = 0.05 m and z = −0.05 m planes, respectively, seen in (b) and (d).

## 3. Results and Discussion

Optical experiments are carried out to verify that the method introduced above can be used to achieve the purpose of focusing multiple curve beams in focusing regions. The experimental arrangement for creating the curve beams is sketched in Figure 5. The optical setup for generating the curve beams is composed of a Liquid Crystal spatial light modulator (SLM), a 4-f filtering system, and a Fourier transform (focusing) lens. A solid-state laser with a wavelength of 532 nm is collimated to plane wave illumination. The SLM (Holoeye Pluto, 8-pixel pitch, 1920 × 1080 resolution) is utilized to address a phase-only CGH. We use the double-phase method [16,17] to encode the complex CGH H(x,y) that is calculated by (5) into a phase-only CGH. It consists of the encoding of the complex function as a hologram into the SLM. The beam modulated by SLM is then projected to the back-aperture of the Fourier transform lens (f = 400 mm) through a 4f optical filtering configuration. A charge coupled device (CCD) camera is placed at the Fourier plane of the focusing lens to record the generated intensity patterns.

Figure 6 shows the experimental reconstruction results consisting of four different curve beams, which clearly indicates that the experiment results are in full agreement with the simulation. From the experimental results shown in Figure 6a,b, four curve beams are focused simultaneously on the focal plane (z = 0), which are consistent with the simulation results in Figure 2, Two arrangements show that the method can adjust the position of the curve as needed. From the experimental results shown in Figure 6c–j, the displacement of curves to the focal plane is set by z1 = −0.05 m, z2 = 0 and z3 = 0.05 m, respectively, and is recorded by moving the CCD back and forth along the z direction. We can also see that the experimental results are consistent with the simulation results in Figure 3 and Figure 4. Focused ring curve and square curve beams can be observed on the original focal plane (z = 0). Therefore, we experimentally verified the feasibility of this method.

In fact, the high intensity gradient along the light curve and the determined phase gradient allow optical vortices to be useful in optical manipulating applications. The beam along an arbitrary curve is generated by a holographic technique in reference [13]. We can generate multiple curve beams at the same time and keep properties such as HIG and phase gradient unchanged. Although Deng et al. [12] proposed an approach for creating multifocal vortices arrays, they only generate ring curves of beams on different regions. In this paper, the optical vortices can be designed in other shapes besides the ring, and the relative position of the Z direction can be adjusted flexibly. Therefore, with this method, the number of laser traps can be increased or decreased as needed. The relative position between different beams can also be flexibly adjusted to meet the needs of different applications. Some applications may require a three-dimensional continuous operation of the particles, so generation of multiple beams along three dimensions curves will be our new direction in future studies.

## 4. Conclusions

A single curve beam with high intensity and phase gradient along an arbitrary curve can be generated, using the holographic technique. More innovatively, we theoretically and experimentally show that optical vortices of different shapes can be focused at different locations on a single plane. In addition, multiple beams can be focused on different regions in the z direction. Furthermore, the shape and position of optical vortices can be freely changed as needed. Unlike conventional optical tweezers, which rely on adjusting the positions of intensity gradients to move objects, the resulted beams can achieve an efficient optical trap. This work is believed to be meaningful and useful in the further development of optical vortices for multitasking optical applications.

## Figures and Tables

**Figure 1 nanomaterials-09-00087-f001:**
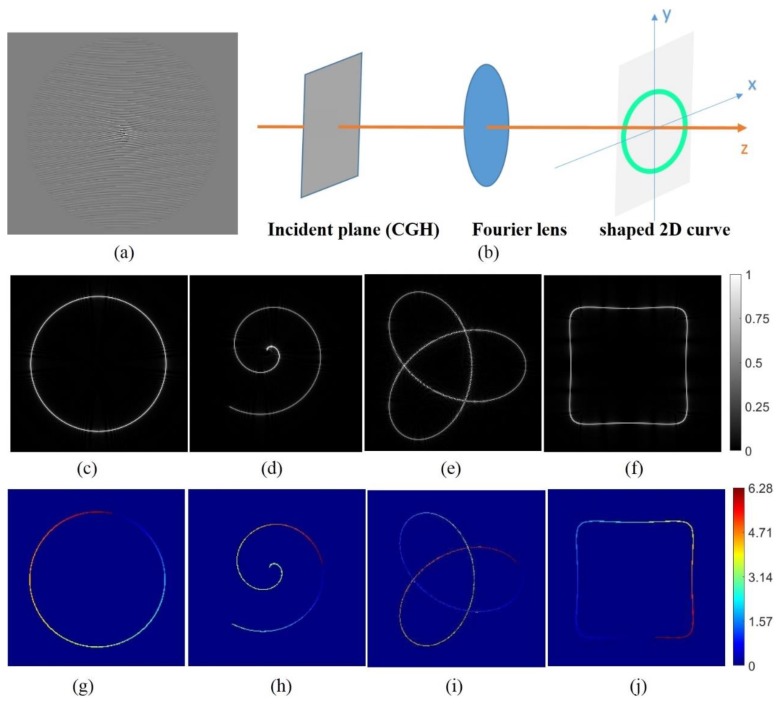
(**a**) Computer-generated hologram (CGH) of a 2D ring curve beam (**b**) Scheme of holographic three-dimensional beam shaping technique. (**c**,**g**) Reconstructed intensity and phase distribution of the ring curve at the focal plane. (**d**,**h**) Reconstructed intensity and phase distribution of the Archimedean spiral at the focal plane. (**e**,**i**) Reconstructed intensity and phase distribution of the trefoil-knotted curve at the focal plane. (**f**,**j**) Reconstructed intensity and phase distribution of the square curve at the focal plane.

**Figure 2 nanomaterials-09-00087-f002:**
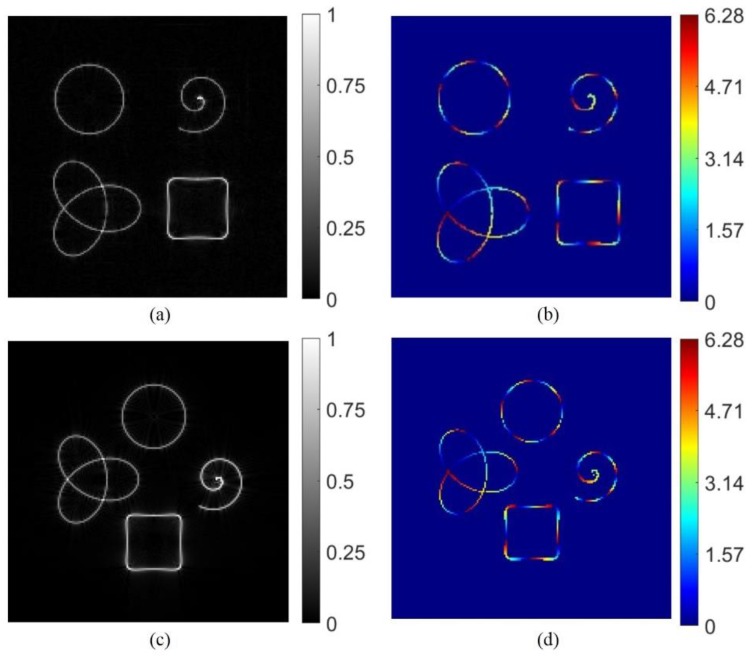
(**a**,**c**) Reconstructed intensity of the multiple light beams at the focal plane. (**b**,**d**) Phase distribution of the multiple light beams at the focal plane.

**Figure 3 nanomaterials-09-00087-f003:**
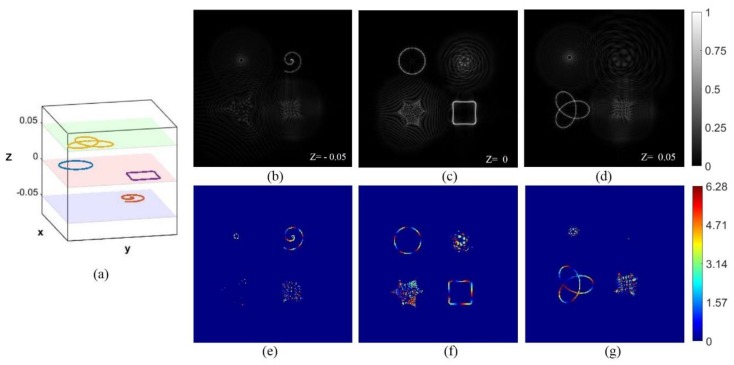
The beams focused on different regions. A three-dimensional schematic of focused beams is seen in (**a**). Intensity distribution of the beams projected in the focal plane (z = 0) seen in (**c**), and the Archimedean spiral and the trefoil-knotted curve are focused on the z = −0.05 m and z = 0.05 m planes, respectively, seen in (**b**,**d**). (**e**–**g**) The phase distribution of the multiple light beams.

**Figure 4 nanomaterials-09-00087-f004:**
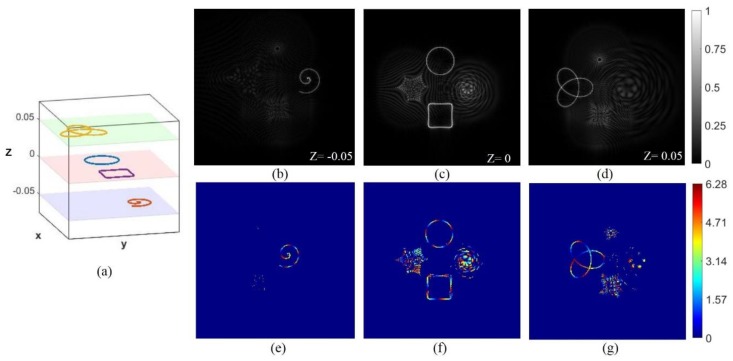
The beams focused on different regions. A three-dimensional schematic of focused beams is seen in (**a**). Intensity distribution of the beams projected in the focal plane (z = 0) seen in (**c**), and the Archimedean spiral and the trefoil-knotted curve are focused on the z = −0.05 m and z = 0.05 m planes, respectively, seen in (**b**,**d**). (**e**–**g**) The phase distribution of the multiple light beams.

**Figure 5 nanomaterials-09-00087-f005:**
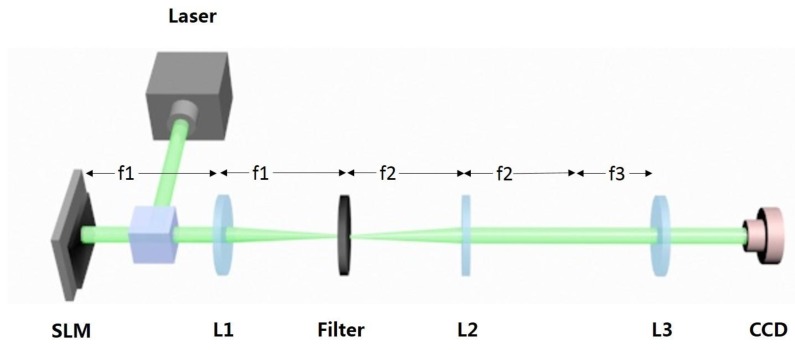
Experimental setup. The hologram is addressed into the SLM, which is illuminated by a collimated laser beam. After the beam passes through lens 1, the desired pattern can be filtered with a diaphragm. Then resulted beams pass through lens 2 and lens 3, and can be captured by the camera.

**Figure 6 nanomaterials-09-00087-f006:**
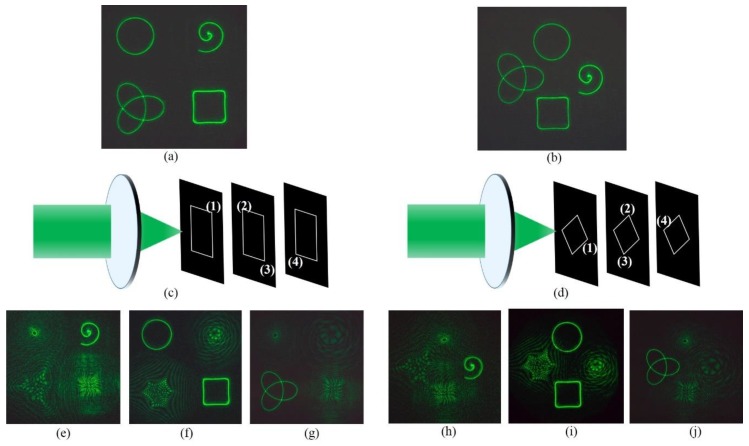
(**a**,**b**) Experimental results. The resulting beams are photographed on the focal plane. (**c**–**j**) Experimental results of the four different curve beams in two different types of three-dimensional layouts, under different analyzer directions.

**Table 1 nanomaterials-09-00087-t001:** Parametric expressions for different curves.

Type of Curve	x_0_(t)	y_0_(t)
ring curve	Rcos(t)	Rsin(t)
Archimedean spiral	−Rtcos(10t)	−Rtsin(10t)
trefoil-knotted curve	Rcos(t) − 2Rcos(2t)	Rsin(t) + 2Rsin(2t)
square curve	−2Rcos(t) + 0.3Rcos(kt)	−2Rsin(t) + 0.3Rsin(kt)

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
