# Peer review of "Simultaneous Generation of Complex Structured Curve Beam"

_nanomaterials, 2019, doi:10.3390/nano9010087_

Reviewer 1 Report

The paper is written in clear way and the results undoubtedly deserve publication in Nanomaterials. I would only like to know if the authors of the paper are planning to improve the method of obtaining the multiple beams in different positions of the focal regions? What are the future prospects of work in this direction? 

Author Response

Comments and Suggestions for Authors:

The paper is written in clear way and the results undoubtedly deserve publication in Nanomaterials. I would only like to know if the authors of the paper are planning to improve the method of obtaining the multiple beams in different positions of the focal regions? What are the future prospects of work in this direction?  

Reply:

Thank you for your comment. For the method of obtaining the multiple beams in different positions of the focal regions, we think it is an efficient and convenient method at present. However, we are not satisfied with generating multiple beams in a single plane. We hope that multiple beams can be generated in different areas of three-dimensional space. In the future, complex structured curve beam can be used in optical micro-manipulation, biomedicine, quantum communication and other applications.

Reviewer 2 Report

In this paper, the authors present a method for simultaneously shaping of multiple beam lattice where the intensity and phase of each individual beam can be prescribed along arbitrary geometric curve. First, the theory and the simulation results are discussed, then the actual experimental results are shown to confirm the validity of their approach. In this approach, different arbitrary curved beams can be simultaneously focussed on different transverse and longitudinal point. I think this paper deserve publication.

In the next line after Eq.(1), "φ and φ" may be "φ and fai(in Eq.(3))''.

Author Response

Comments and Suggestions for Authors:

In this paper, the authors present a method for simultaneously shaping of multiple beam lattice where the intensity and phase of each individual beam can be prescribed along arbitrary geometric curve. First, the theory and the simulation results are discussed, then the actual experimental results are shown to confirm the validity of their approach. In this approach, different arbitrary curved beams can be simultaneously focussed on different transverse and longitudinal point. I think this paper deserve publication.

In the next line after Eq.(1), "φ and φ" may be "φ and fai(in Eq.(3))''.

Reply:

We thank the reviewer for this comment. We have revised it as required and highlighted it clearly in current version.

Reviewer 3 Report

The paper describes a novel technique of generation of spatially shaped optical vortices. The results are important and clearly deserve to be published. The theoretical formalism is well justifies, and the experimental techniques are consistent. I suggest that the authors include a discussion of a recent publication in relation to their work:  D. Sarenac, D. G. Cory, J. Nsofini et al, Generation of a Lattice of Spin-Orbit Beams via Coherent Averaging, Phys. Rev. Lett. 121, 183602 .

After this minor addition, I will be glad to recommend the manuscript for publication.

Author Response

Comments and Suggestions for Authors:

The paper describes a novel technique of generation of spatially shaped optical vortices. The results are important and clearly deserve to be published. The theoretical formalism is well justifies, and the experimental techniques are consistent. I suggest that the authors include a discussion of a recent publication in relation to their work:  D. Sarenac, D. G. Cory, J. Nsofini et al, Generation of a Lattice of Spin-Orbit Beams via Coherent Averaging, Phys. Rev. Lett. 121, 183602 .

After this minor addition, I will be glad to recommend the manuscript for publication.

Reply:

We thank the reviewer for providing the reference. D. Sarenac described a parallel multiplexing technique that produced a beam consisting of a lattice of OAM states coupled to a two-level system, through N sets of Lattice of Optical Vortices prism pairs. We think it is a good method for both electromagnetic and matter-wave beams. We have cited it in our revised article.